# Redundant representations are required to disambiguate simultaneously presented complex stimuli

W. Jeffrey Johnston[1,2]*, David J. Freedman[1,3]

**1** Graduate Program in Computational Neuroscience and the Department of Neurobiology, The University of Chicago, Chicago, Illinois, United States of America, **2** Center for Theoretical Neuroscience and Mortimer B. Zuckerman Mind, Brain and Behavior Institute, Columbia University, New York, New York, United States of America, **3** Neuroscience Institute, The University of Chicago, Chicago, Illinois, United States of America

* wjeffreyjohnston@gmail.com

**Data Availability Statement:** All of the code underlying this work is available at: https://github.com/wj2/assignment-problem.

**Funding:** This work was supported by NIH F31EY029155 (WJJ), NSF 1707398 (WJJ), Simons

## Abstract

A pedestrian crossing a street during rush hour often looks and listens for potential danger. When they hear several different horns, they localize the cars that are honking and decide whether or not they need to modify their motor plan. How does the pedestrian use this auditory information to pick out the corresponding cars in visual space? The integration of distributed representations like these is called the assignment problem, and it must be solved to integrate distinct representations across but also within sensory modalities. Here, we identify and analyze a solution to the assignment problem: the representation of one or more common stimulus features in pairs of relevant brain regions—for example, estimates of the spatial position of cars are represented in both the visual and auditory systems. We characterize how the reliability of this solution depends on different features of the stimulus set (e.g., the size of the set and the complexity of the stimuli) and the details of the split representations (e.g., the precision of each stimulus representation and the amount of overlapping information). Next, we implement this solution in a biologically plausible receptive field code and show how constraints on the number of neurons and spikes used by the code force the brain to navigate a tradeoff between local and catastrophic errors. We show that, when many spikes and neurons are available, representing stimuli from a single sensory modality can be done more reliably across multiple brain regions, despite the risk of assignment errors. Finally, we show that a feedforward neural network can learn the optimal solution to the assignment problem, even when it receives inputs in two distinct representational formats. We also discuss relevant results on assignment errors from the human working memory literature and show that several key predictions of our theory already have support.

## Author summary

Human and animal behavior relies on the integration of distinct sources of information about the same objects in the world—for instance, social behavior requires the correct integration of people with their words, even when multiple people are talking over each

Foundation 542983SPI (WJJ), Gatsby Charitable Foundation GAT3708 (WJJ), NIH R01EY019041 (DJF), CRCNS NIH R01MH115555 (DJF), NSF NCS 1631571(DJF), and a DOD Vannevar Bush Fellowship (DJF). The funders provided salary and other support to both authors as noted above. The funders had no role in study design, data collection and analysis, decision to publish, or preparation of the manuscript.

**Competing interests:** The authors have declared that no competing interests exist.

other. We formalize this integration process and show that it relies on at least partial redundancy between these different sources of information. In the case of integrating vocalizations with their source, this redundancy could be provided by the distinct representations of spatial position in the visual and auditory systems. Then, we show that the necessity of this integration process produces a trade-off between the representation of redundant information (for reliable integration) and the representation of non-redundant information (which is to be integrated), with implications for modular organization in the brain. Finally, we show that a simple feedforward neural network can integrate as reliably as predicted by our theory—as well as make predictions from our theory that can be tested in neural data. Overall, this work provides insight into how the brain makes sense of its distributed and sometimes distinct representations of the world.

## 1 Introduction

Coherent behavior in complex natural environments requires extensive and reliable integration of different forms of information about the world. For instance, a pedestrian crossing a crosswalk during rush hour attends to the flow of traffic around the intersection—if they hear honking, it is important to quickly localize that honking to cars in visual space and decide whether they need to change their motor plan. While navigating cluttered multisensory environments like this one often appears effortless for humans and other animals, it requires two highly non-trivial computations: Object segmentation and representation assignment. For object segmentation, the continuous sensory world must be segmented into distinct objects— that is, each car has to be recognized as a distinct entity, separated from both the other cars and background objects, such as buildings. The primitive rules that humans and other animals use to segment objects have been studied extensively as part of gestalt psychology [1] and related subfields [2, 3], as well as in machine learning [4, 5]. In the brain, this segmentation process is thought to be represented by border ownership cells, such as those observed in the primate visual system [6, 7]. However, even once this difficult object segmentation problem is solved, the resulting object representations are often still incomplete: An object representation in one brain region contains only a part of all the information about that object that is present in the entire brain. The integration of these distributed representations is referred to as representation assignment [8]—and is analogous to the extensively studied assignment problem from combinatorial optimization [9]. In the example above, the representations of the cars from the visual system need to be integrated with the representations of the cars from the auditory system, as both sensory systems provide information that is necessary to guide the pedestrian's motor plan.

The integration of these distinct, parallel representations of the world has been previously studied in three principle ways. First, the integration of distinct features within the visual system has been studied in the context of feature integration theory [10, 11]. In feature integration theory, spatial attention is used to bind features together due to their spatial proximity. Spatial attention is deployed to different locations in sequence and only a single object is bound at a time. However, experimental work has shown that errors can occur in this integration process [12, 13], in which some of the features from two different locations are assigned to each other. We refer to these kinds of errors as an assignment errors, though they are also referred to as illusory conjunctions, swap errors, and misbinding errors. Assignment errors produce representations—such as a barking cat and meowing dog, see Fig 1A—that can be particularly catastrophic for coherent behavior. While feature integration theory

 

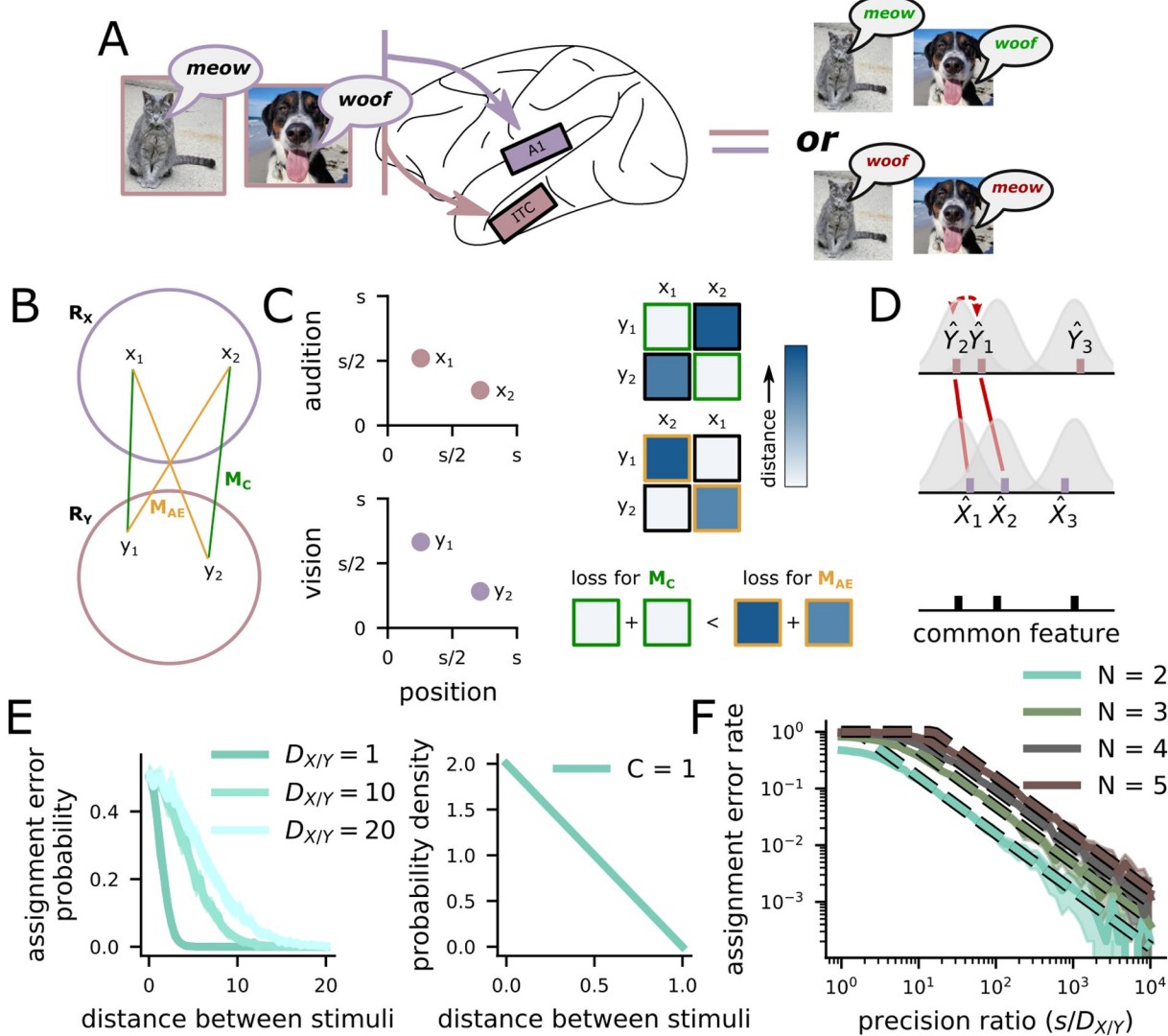

**Fig 1. The assignment problem arises from distributed representations of multiple stimuli. A** A schematic of the assignment problem. The brain receives both visual and auditory information about a dog and a cat, this information is initially separated in the brain. When combining the two sets of representations (auditory and visual), there are two possible integrations, one that correctly reconstructs a barking dog and meowing cat (left) and the other that incorrectly constructs a barking cat and meowing dog (right). The right is an example of an assignment error. **B** Schematic of the assignment process as selecting a mapping between two sets of representations. **C** (left) The assignment problem can be solved by the representation of a common stimulus feature in both brain regions. In our example, the auditory (top) and visual (bottom) representations can be integrated through a shared representation of azimuthal position (the shared x-axis). (right) The $N \times N$ cost matrix (here, each element is given by the distance in the shared feature space across representations from the two regions) for the correct (top) and incorrect (middle) mappings. The most likely assignment minimizes the trace of the matrix (bottom). **D** Assignment errors occur when the estimate of the common feature value for two stimuli cross over each other in one region, but not the other (middle, red arrow). The distribution around each point is the local MSE ($D_X$ and $D_Y$). **E** (left) The probability that this crossing over occurs is high for nearby stimuli and low for distant stimuli, and increasing estimator variance makes assignment errors more likely at all distances. (right) For stimuli that are uniformly distributed in the full feature space, the distance between pairs of stimuli follows a triangular distribution with one commonly represented feature ($C = 1$). **F** The overall assignment error rate is the product of the two functions on the left—the assignment error probability at each distance weighted by the probability that there is a pair of stimuli at that distance. Dashed line: theory from Eq 3; solid line: simulations.

provides a qualitative description of the assignment process using a common representation of spatial position to link different feature representations together, it does not provide a mechanistic explanation of how assignment errors depend on the qualities of the underlying neural representations.

Second, related to feature integration theory, the assignment problem has been studied as part of the literature on working memory capacity. Here, participants are asked to remember an array of multi-feature stimuli (e.g., oriented and colored bars at different spatial locations) across a delay period, before being cued by a single feature (e.g., position) to report one of the other features (e.g., color) [14–17]. In this setting, the cue must be assigned to the correct representation from the set of remembered stimuli. Here, the assignment problem is solved across two distinct time points rather than two distinct brain regions, but the underlying principles are the same. This literature shows that assignment errors increase with the number of stimuli that need to be remembered [15, 16] and that they become more likely when remembered stimuli are close together in the feature that is used to cue a stimulus for report (from above, spatial position) [16, 17]. The assignment error rate also depends on the precision of the representation of the cued feature [17]. Further, a neural model has been developed to explain these behavioral findings, which links the assignment error rate to neural population codes [16]. Our work builds on the theoretical work from the time-assignment setting by considering multiple common (i.e., cue) features [18] (but see [19]), as well as through its focus on assignment across (and within, as in feature integration theory) distinct brain regions at a single time point.

Third, representation assignment has also been studied in the context of multisensory integration [8, 20–23]. In this literature, both experiment and theory have primarily focused on the integration of multisensory representations of a single stimulus. This work has demonstrated that shared information is crucial to reliable integration. In particular, integration of auditory and visual information appears to rely on the shared representation of azimuthal position derived from both sensory systems [21]. That is, whether or not an auditory and visual stimulus are integrated depends on the variance in estimates of azimuthal position from both sensory systems [21]. We refer to the variance of estimates from these individual representations as the local variance.

Here, we extend this analysis to sets of multiple objects and develop a general framework that demonstrates how the likelihood of assignment errors depends on the number of common features that are represented across multiple brain regions (such as azimuthal position) and the local variance of those feature representations. We show that there is a tradeoff between highly redundant representations that produce low assignment error rates, but higher local variance, and less redundant representations that risk more assignment errors but have generally lower local variance. To make this tradeoff concrete, we analyze a receptive field model of neural representations, similar to those found in many sensory areas [24–28] and related to models explored in the working memory literature [16]. With this model and when keeping constant the total metabolic resources (i.e., number of neurons and spikes) used by the model, we show that the lowest total error (combined local variance and assignment errors) can sometimes be achieved by systems that split into distinct modules, despite the fact that such systems risk assignment errors. Thus, our framework provides a potential explanation for the extensive modularity and parallel processing thought to exist in the primate visual system [29–32] and other sensory systems [33–36]. Finally, we demonstrate that the assignment problem can be solved optimally by a feedforward neural network. In the feedforward network, we also show how the integrated stimuli can be reliably represented: through nonlinearly mixed representations of the integrated features, which follows from previous work on representations of multiple stimuli [16, 37, 38]. Finally, we discuss predictions for neural data, and link our results in more detail to behavioral results from the human working memory literature. Overall, this work demonstrates a general solution to the representation assignment problem, which arises whenever a distributed neural system represents multiple stimuli—within single brain regions, across parallel brain regions, and across different time points.

## 2 Results

### 2.1 Redundant information is necessary to solve the assignment problem

Here, we investigate the assignment problem that arises when each of $N$ stimuli have representations that are split across two brain regions, $R_X$ and $R_Y$—in combinatorial optimization, this is referred to as the balanced assignment problem [9]. We want to correctly integrate the distributed representations of these $N$ stimuli, which we formalize as finding the correct one-to-one mapping between the representations in $R_X$ and the representations in $R_Y$. To give an example: a dog and a cat will be represented in at least two distinct regions of the brain, an auditory region $R_X$ and a visual region $R_Y$ (Fig 1a). The set of representations in $R_X$, referred to as $\hat{X}$, will consist of representations of each of the animals' vocalizations—similarly, $\hat{Y}$ in $R_Y$ consists of representations of each animals' visual features (see Methods in Definition of the representations for more details). Many behaviors rely on integrated information about both the visual features and vocalizations of the animals (for instance, when deciding which animal to approach, it is useful to know whether they are barking or meowing).

So, we want to choose the correct one-to-one map $M$ between $\hat{X}$ and $\hat{Y}$ (Fig 1b). We show that if $\hat{X}$ and $\hat{Y}$ are independent, then all maps are equally likely (see Methods in Definition of assignment errors). In this case, assignment errors will occur with probability $1 - 1/N!$. Thus, redundancy between the representations in $R_X$ and $R_Y$ is necessary to reliably solve the assignment problem. More concretely, the case where $\hat{X}$ and $\hat{Y}$ are independent would be like the case where, in a working memory study similar to the ones discussed in the Introduction, we ask the participant to remember the color and orientation of a collection of bars, then cue them with a spatial frequency. Without additional instruction or extensive learning, the participant could do no better than chance—just as our system here could solve the assignment problem no more reliably than by guessing.

The required redundancy can take many forms. We will focus on the case where the redundancy manifests as a linear dependence between some of the features represented in $R_X$ and some of the features represented in $R_Y$. In the working memory task example, this is similar to remembering both the color and orientation of each stimulus in an array. Then, being asked to report the color of the stimulus that had a particular orientation. Our analysis can also be extended to some kinds of nonlinear mappings between between the features in $R_X$ and $R_Y$ (see Fig B in S1 Text).

One specific kind of nonlinear redundancy between the two regions is prior information about which stimuli are more or less likely. For example, someone familiar with cats and dogs could infer that a meowing noise likely belongs to the cat they see in front of them, while the barking likely belongs to the dog. While such inferences are likely extremely important for behavior, they require the prior information to be learned over experience with these particular stimuli. We focus our analyses here on solving the assignment problem without this learning process (as for novel stimuli) and where such differences in prior likelihood are not available (e.g., assigning two different barks to two different novel dogs).

Finally, throughout this work we organize our discussion using the concept of distinct brain regions, by which we mean distinct populations of neurons that may share information with each other through anatomical connections. We primarily use this concept for simplicity of exposition. In general, all of the same principles apply when different features (or combinations of features) are represented in distinct subspaces of the same neural population activity. That is, the assignment problem will also need to be solved to integrate across these distinct subspaces, even though the subspaces do not correspond to distinct populations of neurons.

## 2.2 The assignment error rate depends on the precision of the common feature representation

Here, we study redundancy between $R_X$ and $R_Y$ that results from the representation of one or more of the same stimulus features (e.g., azimuthal position in both the visual and auditory systems, Fig 1c, left). These $C$ common, or linearly related, features define a $C$-dimensional common feature space. Importantly, we do not yet make any assumptions about the representational format of these features—and while the common features are assumed to be the same (or, at least, correlated) across the two regions, the format of their representation need not be.

Then, following the combinatorial assignment problem, we show that the most likely mapping is the one that minimizes the sum squared distance between integrated pairs in this common feature space (Fig 1c, right: top matrix is the the correct mapping, $M_C$, bottom matrix is the incorrect mapping $M_{AE}$, the sum of distances between integrated pairs in the two mappings are shown below). If the estimates of the common features in both regions are exactly correct, then this sum will be zero—an exact matching (Fig 1c). However, in general, we assume that each feature represented in $R_X$ is decoded with Gaussian-distributed local variance $D_X$ (Fig 1d). Further, we show that assignment errors occur precisely when the representation of one object crosses over the representation of another object in one region but not the other (Fig 1d, red lines). For two objects at distance $\delta$ from each other in the common feature space space, the probability of an assignment error between those two stimuli is approximated by

$$F(\delta) \approx Q\left(\frac{-\delta}{\sqrt{2D_X}}\right) + Q\left(\frac{-\delta}{\sqrt{2D_Y}}\right) \qquad (1)$$

where $Q(x)$ is the cumulative density function for the standard normal distribution (Fig 1e, left)—for the full expression and derivation, see Methods in Probability of assignment errors.

The overall probability of an assignment error also relies on the probability that two objects are at distance $\delta$ from each other in the common space $P_C(\delta)$ (Fig 1e, right for $C = 1$) and on the number of stimuli $N$. Incorporating these, we show that the overall probability of an assignment error is upper bounded by,

$$AE_C \leq \binom{N}{2} \int d\delta\, P_C(\delta)\, F(\delta) \qquad (2)$$

For one overlapping stimulus feature ($C = 1$) and for stimuli that are uniformly distributed in feature space, we show that,

$$AE_1 \approx \binom{N}{2} \frac{2\sqrt{D_X + D_Y}}{s\sqrt{\pi}} \qquad (3)$$

where $s$ is the size of the feature space for the commonly represented feature. This approximation (dashed lines, Fig 1f) closely matches the empirical assignment error rate (solid lines) across different numbers of stimuli (different colors; see Methods in Assignment error rate approximation for $C = 1$ for a full derivation of this expression). Now, using this formalization, we show that the assignment error rate decreases with additional commonly represented features $C > 1$ in the next section. We also show that the assignment error rate increases if the features are represented with asymmetric variance across the two brain regions—i.e., if $D_X \neq D_Y$ (see Fig A in S1 Text).

## 2.3 Additional commonly represented features decrease the assignment error rate

While distinct sensory systems tend to have fixed amounts of overlapping information, such as a common estimate of azimuthal position across the auditory and visual systems (which has been shown to be crucial for single stimulus integration [21]), within a single sensory system information is distributed across multiple brain regions, and those brain regions can represent variable amounts of overlapping information about the stimuli. In our framework, we show that increasing the number of commonly represented stimulus features across the two brain regions $C$ decreases the assignment error rate.

In particular, increasing the number of commonly represented features $C$ increases the average distance between pairs of points in the common feature space (Fig 2a). This change is captured by changes in the $p_C(\delta)$ term in Eq 2. Increasing the number of commonly represented features moves the probability mass of this distribution away from zero (Fig 2b). The increase in the number of overlapping features from $C$ to $C + 1$ leads to an orders of magnitude decrease of the assignment error rate at the mean distance (Fig 2c). As a result of this shift to the right in $p_C(\delta)$ and the fact that Eq 1 is strictly (and exponentially) decreasing in $\delta$, the integral in Eq 2 will also be strictly decreasing as $C$ is increased. Thus, increasing $C$ decreases the overall probability of assignment errors when holding local variance constant (Fig 2d).

However, increasing $C$ while holding local variance constant comes with a significant cost. In particular, increasing the number of commonly represented stimulus features also increases the redundancy in information represented by the two brain regions. As discussed above, it is precisely this redundancy that allows the assignment problem to be solved (and solved more reliably for more overlapping features). Yet, in neural systems with limited metabolic resources, this redundancy reduces the efficiency of neural representations. We show that the redundancy between the two regions depends on both the number of overlapping features and the local variance (Fig 2e),

$$
\begin{aligned}
R \quad &= I(X; Y) = H(X) - H(X|Y) \\
&= \frac{C}{2} \log \frac{s^2}{D_X + D_Y}
\end{aligned}
$$

where $I(.; .)$ is the mutual information and $H(.)$ is the Shannon entropy, and $s$ is the extent of the overlapping features (see Methods in Calculating the redundancy between representations for more details). Thus, as the assignment error rate is reduced by increased $C$, the redundancy of the representation is increased—we make the consequences of this explicit below.

Further, we show that a similar tradeoff holds for asymmetric feature representations (Fig A in S1 Text). On one hand, the common features can be represented with the same precision in both regions. This is the maximally redundant representation for a given $C$, and therefore leads to the lowest assignment error rate. On the other hand, the common feature can be represented asymmetrically (with high variance in one region and low variance in the other). This is a less redundant representation, and is associated with a higher assignment error rate.

## 2.4 Redundancy reduces the precision of the neural code

Now, we make our theory more concrete by making an assumption about the format of the neural representations in $R_X$ and $R_Y$. This assumption allows us to quantify how the assignment error rate and local variance both depend on the metabolic energy available to the code, in the form of spikes and neurons. To begin, we analyze the errors made by a population of neurons with randomly positioned Gaussian receptive field (RF) responses (Fig 3a, top). This

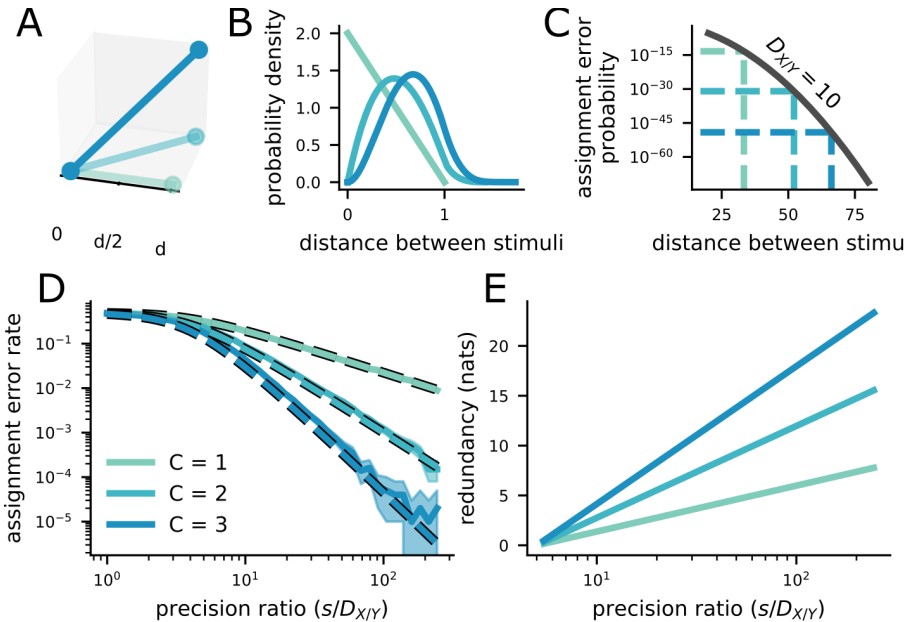

**Fig 2. Increasing the number of commonly represented features decreases the assignment error rate, but increases the level of redundancy between the representations.** The color legend is the same throughout the plot, and provided in **D**. **A** Schematic showing that the distance between two points increases with the dimensionality of the space that they are in. **B** The distribution of distances between two uniformly distributed points in a space of one, two, and three dimensions. The distribution shifts to the right (toward larger distances) as the dimensionality of the space increases. This is true for points with any distribution that has non-zero variance. **C** At the average distance between two points (dashed lines), the probability of an assignment error decreases by orders of magnitude as the dimensionality of the commonly represented space increases, without changing the estimator variance for the representations $D_{XY} = 10$. **D** The overall assignment error rate also decreases by orders of magnitude as the dimensionality of the commonly represented feature space increases, while holding $D_{XY}$ constant. The difference becomes even larger as the precision ratio increases. **E** The redundancy between representations $\hat{X}$ and $\hat{Y}$ also increases as the dimensionality of the commonly represented feature space increases—again, the difference is increased at larger precision ratios. Thus, the assignment error rate is driven down at the cost of additional redundancy.

kind of neural code is thought to be used for a variety of different sensory features [24–28], including spatial position in the visual system (but, interestingly, not in the mammalian auditory system [39–42]). While we assume that $R_X$ and $R_Y$ both use the same representational format, this is not required by our theory of assignment errors—and we explore the case in which $R_X$ and $R_Y$ have different representational formats below.

Recent work has shown that RF codes make two kinds of errors [43]. The first kind of error is local, and is equivalent to the local variance we have discussed so far. In this case, the decoder maps the noisy response to a nearby position on the stimulus representation manifold (Fig 3a, bottom, "local error"). This local variance is captured by the Fisher information of the code [43–49]. The second kind of error is non-local, and is referred to as a threshold (or catastrophic) error [43, 50]. Catastrophic errors occur when the noisy response is mapped to a distant region on the stimulus manifold, due to large noise that points toward one of these regions of response space (Fig 3a, bottom, "threshold error"). In a noisy system, every response will give rise to some local variance (Fig 3b, "local"); threshold errors only occur for a subset of responses, but their magnitude has the same order as the size of the stimulus space (Fig 3b, "threshold").

These two kinds of error are in tension with each other. To see this, we begin by deriving the average Fisher information of a random RF code with a particular number of neurons and

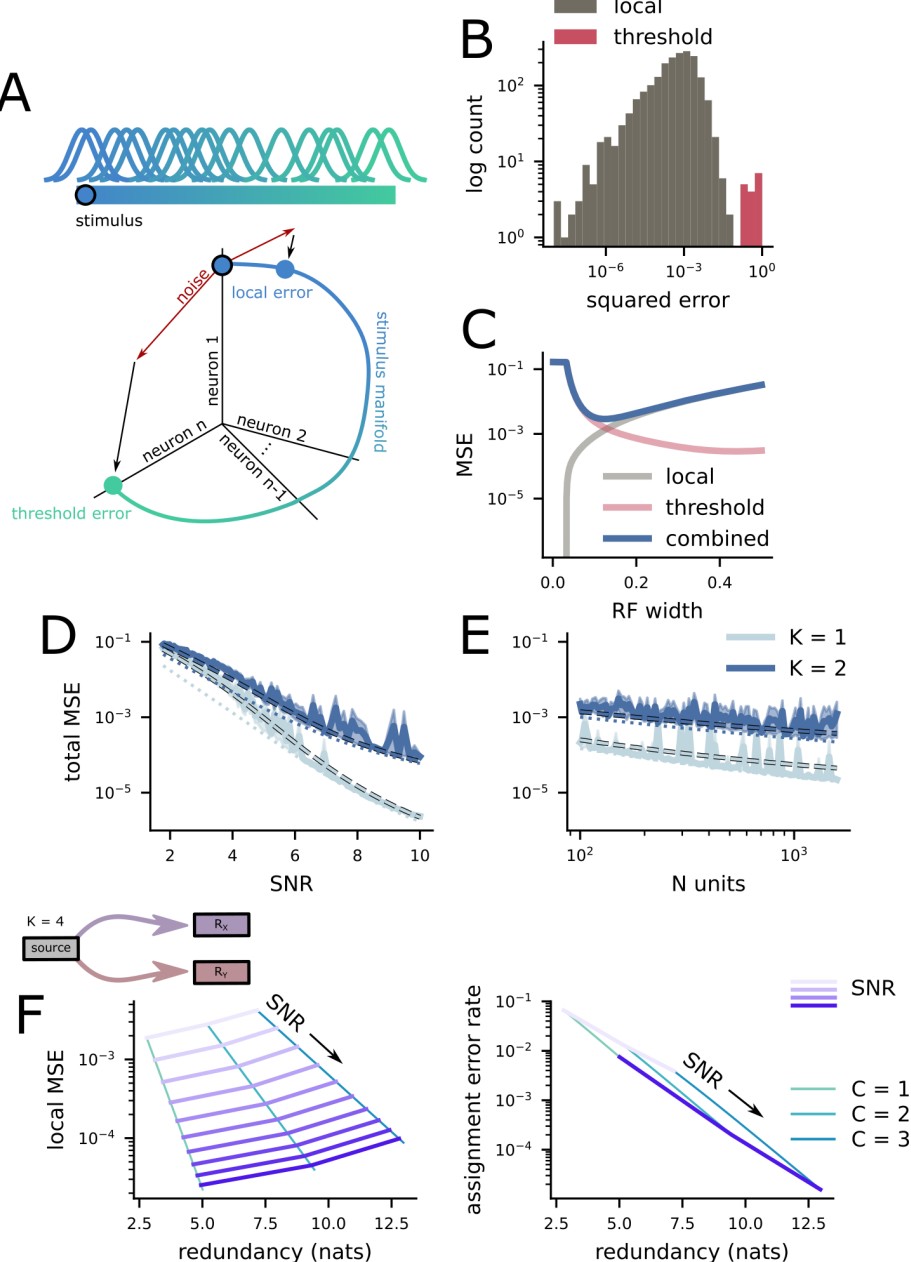

**Fig 3. The efficiency of random receptive field codes. A** A random Gaussian receptive field population. The width of the RFs are chosen to minimize the total MSE of the population. **B** A histogram of errors made by an example RF code. While the majority of errors are local (left, grey), a small proportion of errors are non-local threshold errors (right, red). **C** The two kinds of errors made by RF codes. Threshold errors dominate at small RF sizes, while local errors dominate elsewhere. We choose the RF size for our codes based on the combined error rate. **D** The dependence of total MSE on population SNR for fixed population size. The solid line is from simulations, the dashed line is the prediction of our theory, and the dotted line is local MSE only. **E** The dependence of total MSE on population size for fixed population SNR. The different line types are the same as **D**. **F** (above) Schematic of the situation being investigated. A $K = 4$ dimensional stimulus is represented across two distinct populations of neurons $R_X$ and $R_Y$. Each population of neurons has 10, 000 units and an SNR varying between 5 and 10. (left) The average local MSE across the two codes and the redundancy across the two regions of the code schematized above, shown for different numbers of overlapping features $C$. The grey lines connect points with the same SNR across the different overlap levels. (right) The same as on the left, but for the assignment error rate.

a particular average level of activity across the whole population (i.e., a budget of metabolic energy to be used for spiking),

$$
\begin{aligned}
\text{FI} \quad &\approx \frac{V}{\sigma_N^2 w^2} \frac{\frac{1}{2}\sqrt{\pi} - w}{\sqrt{\pi} - w} \\
&\approx \frac{V}{2\sigma_N^2 w^2} \\
&= \frac{\text{SNR}^2}{2w^2}
\end{aligned}
\tag{4}
$$

where $V$ is the sum squared activity across the population (i.e., the average response energy), $\sigma_N$ is the standard deviation of the Gaussian noise added to the responses, $\text{SNR}^2 = V/\sigma_N^2$ is the population signal-to-noise ratio, and $w$ is the width of the receptive fields, which is assumed to be small (see Methods in Random gaussian receptive field codes for the full derivation). We can see that the Fisher information increases when $V$ increases or when $w$ decreases (Fig 3c, local). However, the total error of the code is given by,

$$
\text{Err} \approx (1 - p_{\text{thr}}) \frac{1}{\text{FI}} + p_{\text{thr}} \frac{1}{6}
$$

where $p_{\text{thr}}$ is the probability of a threshold error, and this probability increases as $w$ decreases (Fig 3c, threshold). All else held equal, decreasing the width of the RFs in the code will reduce the local variance of the code (Fig 3c, local), while simultaneously increasing the probability of threshold errors (Fig 3c, threshold). In the following, we choose the RF width that minimizes the total error of the code (Fig 3c, combined; see Methods in Random gaussian receptive field codes for more details).

Using this framework, we show how this total error scales with both population signal-to-noise ratio ($\text{SNR} = \sqrt{V}/\sigma_N$) and population size. In particular, we show that, for high SNRs, the total error is explained primarily by the local variance (Fig 3d, solid vs dotted line). This results from the fact that the threshold error rate decreases exponentially with SNR (Methods in Random gaussian receptive field codes). Second, we show that increasing the population size while holding the SNR constant leads to a smaller decrease in total error (Fig 3e). That is, while increasing the population size pushes smaller RFs to be optimal, thereby decreasing the local variance, the tension between local and threshold errors causes the decrease in optimal RF size to be small. We also show that, as expected, requiring the code to represent $K = 2$ instead of $K = 1$ features causes higher error rates (Fig 3d and 3E, different lines).

Next, we link these RF codes to our theory of assignment errors. The local variance of the RF code is equivalent to the local variance in our framework. However, the threshold errors have no clear analogue in our current theory. Since the inferred stimulus produced from a threshold error is uniformly distributed across the whole stimulus space, they are extremely disruptive to assignment: The optimal strategy is not to integrate the threshold error representation during the assignment process at all. This unbalanced version of the assignment problem has been studied in combinatorics [9], but is beyond the scope of the current work. However, threshold errors are typically unlikely once the optimal RF width has been chosen. In particular, we find that codes with non-negligible threshold error rates also tend to have high total error. To proceed, we exclude codes with high total error, both due to the issues described for threshold errors and because our analytic calculation for the assignment error rate assumes local variance that is small relative to the size of the stimulus space.

Finally, we analyze the case of $K = 4$ total features represented across two different brain regions, with $C = 1$, 2, and 3 overlapping features. We show that the increase in overlapping

features leads to an increase in both the redundancy between the two representations and the local variance of the code (Fig 3f, left, purple lines show constant SNR). Then, we show that, as anticipated, the increase in the number of overlapping features also leads to a decrease in the assignment error rate at these higher redundancy levels (Fig 3f, right). Thus, this modeling approach allows us to quantify the consequences of different levels of redundancy in a biologically plausible neural code.

## 2.5 Modularity has benefits for encoding high-dimensional stimuli

So far, to motivate the importance of the assignment problem, we have focused on the case of stimulus features that are encoded by separate brain regions as a consequence of being received by distinct sensory systems—such as the pitch of a sound and the spatial frequency of an image. In this case, modularity is unavoidable and the assignment problem must be solved to integrate the disparate percepts. However, there is extensive evidence for modularity in the brain even within single sensory systems (e.g., [30], though these modules are at least partially redundant [51]). In our framework, this comes with a clear drawback when decisions have to be made based on combinations of features represented in different regions: Making those decisions reliably requires the assignment problem to be solved (e.g., identifying which of several cats just hissed or the car in a line of traffic that just honked), and this means that those brain regions must have some redundancy with each other.

Here, we contrast receptive field (RF) codes for relatively high-dimensional stimuli ($K = 8$) that are distributed across one, two, or three brain regions (Fig 4a). For example, a two-region code for the $K = 8$-dimensional stimulus could encode $K_{X,Y} = 4$ features in each region. However, such a code would make assignment errors at chance levels when presented with multiple stimuli. Thus, we consider codes which represent at least one redundant feature for every two brain regions (e.g., one region could represent four features and the other five, as in Fig 4a, middle). In every case where we compare multi-region codes to each other, the total code (that is, summing across brain regions) is constrained to have the same number of neurons and the same total population power.

Within each region, we use the random RF codes described in the previous section. We use the local variance of each region code (Fig 4b, top) to compute the assignment error rate when integrating across regions (Fig 4b, middle), as developed earlier in the paper. Together, these lead to a total error that accounts for local variance and catastrophic assignment errors (Fig 4b, bottom). Codes with total error that exceeds half the size of the stimulus space are excluded from this analysis for the reasons discussed above.

We show that representing the $K$-dimensional stimuli in multiple brain regions can produce representations with lower total error, even when accounting for assignment errors. As a result, this work provides a justification for the modularity observed in the brain from the perspective of reliable and efficient neural codes. For stimuli with $K = 8$ features and smaller neural population sizes or population SNRs, then the lowest total error is achieved by a single region code (Fig 4c, bottom and left). In this case, avoiding catastrophic assignment errors is crucial. However, for larger neural population sizes and population SNRs, codes that distribute the representation over two or even three regions can provide lower total error than the single region code (Fig 4f, top right).

This results from a combination of two factors: First, the multi-region codes must represent redundant information to reliably solve the assignment problem. Thus, they have a disadvantage compared to the single region code, which need not represent any redundant information. However, second, lower-dimensional RF codes tend to produce smaller errors than higher-dimensional codes, especially very high-dimensional codes. In particular, the local

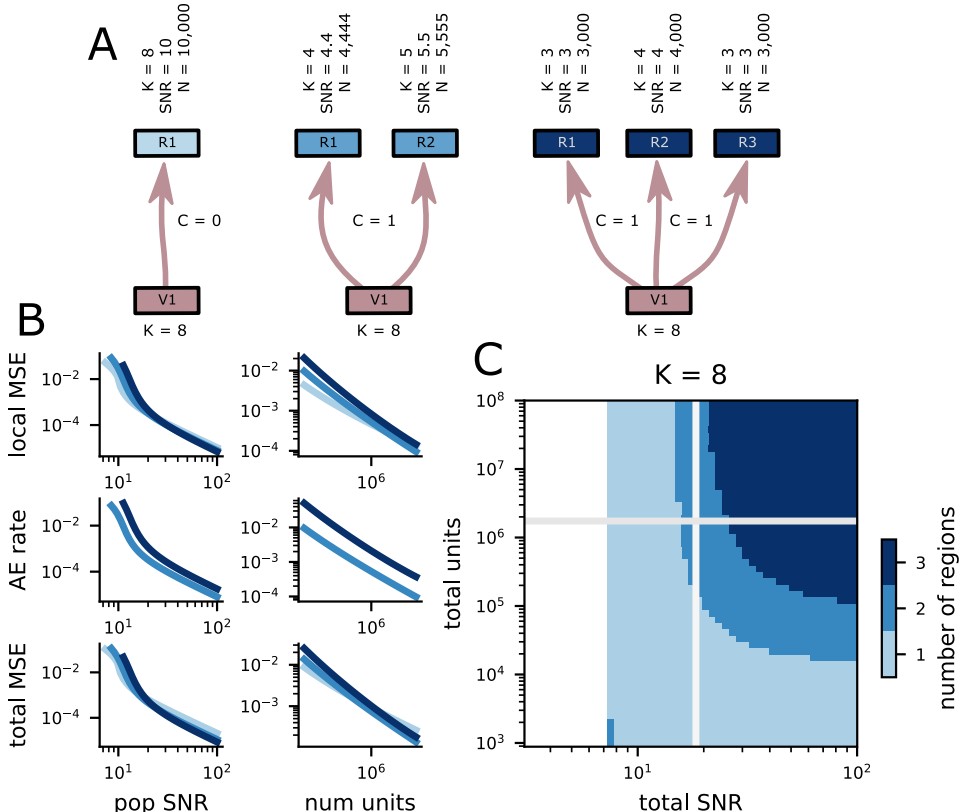

**Fig 4. Multiple-region representations minimize error in many conditions. A** Schematic of three different possibilities for representing a $K = 8$-dimensional stimulus. The features can all be represented in a single population (left); they can be distributed across two populations (middle); they can be distributed across three populations (right). The two and three population codes must represent some features redundantly to avoid assignment errors. **B** (top) The scaling of local MSE for the three codes above with population SNR (left) and the total number of units (right). The colors are the same as in **A** and **C**. (middle) The same as top but showing the scaling of the assignment error rate. (bottom) The same as top but showing the scaling of the total MSE. **C** The number of regions that minimizes the total distortion of the representation of $K = 8$ features for different numbers of neural units and population SNR. The white parts of the plot are when no representational scheme achieved total MSE $< .5$.

variance of an RF code depends on RF width, and not directly on dimensionality (Eq 4). Thus, if low- and high-dimensional codes have the same RF width, then they could, in principle, have the same local variance. In practice, though, the high-dimensional code will require exponentially more neurons than the low-dimensional code to achieve full coverage of the stimulus space at that width (and to control threshold errors). In our setting, the single region, higher-dimensional codes do have more neurons, but only by a factor of two or three—not enough to achieve equivalent local variance to the lower-dimensional RF codes. Thus, multi-region codes begin to outperform single-region codes when the gains from their inherent efficiency become larger than the losses from their required redundancy.

We also observe that the difference in total error between these different coding strategies is relatively small (Fig 4b, bottom). While our current theory does not provide a guarantee of this in all cases, it points to an interesting potential invariance: The increase in redundancy required to represent a set of stimuli across multiple regions appears to be largely compensated for by the increase in efficiency of the constituent codes. Thus, many different coding strategies appear to provide similar performance.

Finally, while we assume that the features are interchangeable with each other, this is often not the case. In general, we believe that the assignment problem is solved to make decisions that require the linking of two or more stimulus features (e.g., asking what the value of feature *X* is for stimuli with a particular value of feature *Y*, as in the human working memory studies discussed above). So, one structural way to reduce the likelihood of assignment errors is to instead represent features *X* and *Y* in the same region, while representing the features that are relevant to a different decision in another region. The primate dorsal and ventral visual streams have been argued to follow this form: where the dorsal stream is thought to underly the planning of visually guided actions and the ventral stream is thought to underly perceptual decisions [31, 52]. However, a single division is unlikely to work for all decisions—and, in some cases, this simply shifts the need of solving the assignment problem to an earlier stage of processing. Thus, we believe that the simplified scenario we have analyzed here still provides a useful intuition.

## 2.6 Feedforward neural networks can solve the assignment problem

In the previous section, we analyzed the best-case assignment error rate for populations of neurons with randomly positioned RFs. However, there is no guarantee that a readout from the neural population will be able to achieve this best-case error rate without extensive and potentially metabolically expensive computations. In particular, solutions to the assignment problem used in combinatorial optimization are often framed as linear programming problems and require operations that have no clear neural analog [9].

Here, we focus on the case with two stimuli that are described by three total features ($K = 3$). The stimuli are initially represented by two distinct populations $R_X$ and $R_Y$—each of which encode two features (Fig 5a, left). For both populations, one feature is unique to that population and the other is common across both populations ($C = 1$). We train a feedforward neural network model (Fig 5b) to take noisy input representations (Fig 5a, right) from $R_X$ and $R_Y$ and produce a representation of the two unique features from $R_X$ and $R_Y$ in an output population, which is also structured as a random RF code (Fig 5a, right). To do this reliably, the network must solve the assignment problem.

First, we train networks to perform this integration (of the two inputs) and marginalization (of the common feature) directly, without a hidden layer (Fig 5b, only $R_X$, $R_Y$, and "output"). Then, we quantify whether the activity in the output network is more similar to the correctly or incorrectly assigned representation (Fig 5d). The network without any hidden layers does not learn to assign the stimuli correctly (Fig 5d, end-to-end learning with no integration layer).

This implies that representations of features from multiple sensory modalities (e.g., the expression of a speaker and their words) may need to be derived from higher-dimensional representations that also incorporate the linking, common feature (e.g., estimates of spatial positions in the previous case). However, this does not imply that all of the features of an object must be represented at once: only those that need to combined, along with however many common features are required to achieve reliable assignment. This two-stage computation (a dimensionality expansion followed by a dimensionality reduction) has been shown to useful for other computations, such as representation frame-of-reference recoding [20].

Second, we show that networks trained to first reconstruct the three dimensional stimulus representations (Fig 5c) and then marginalize out the common feature achieve the minimal assignment error rate, while networks trained directly to construct the marginalized representations achieve low, but not minimal assignment error rates (Fig 5e, end-to-end learning compared to integration learning). While full understanding of this performance difference would

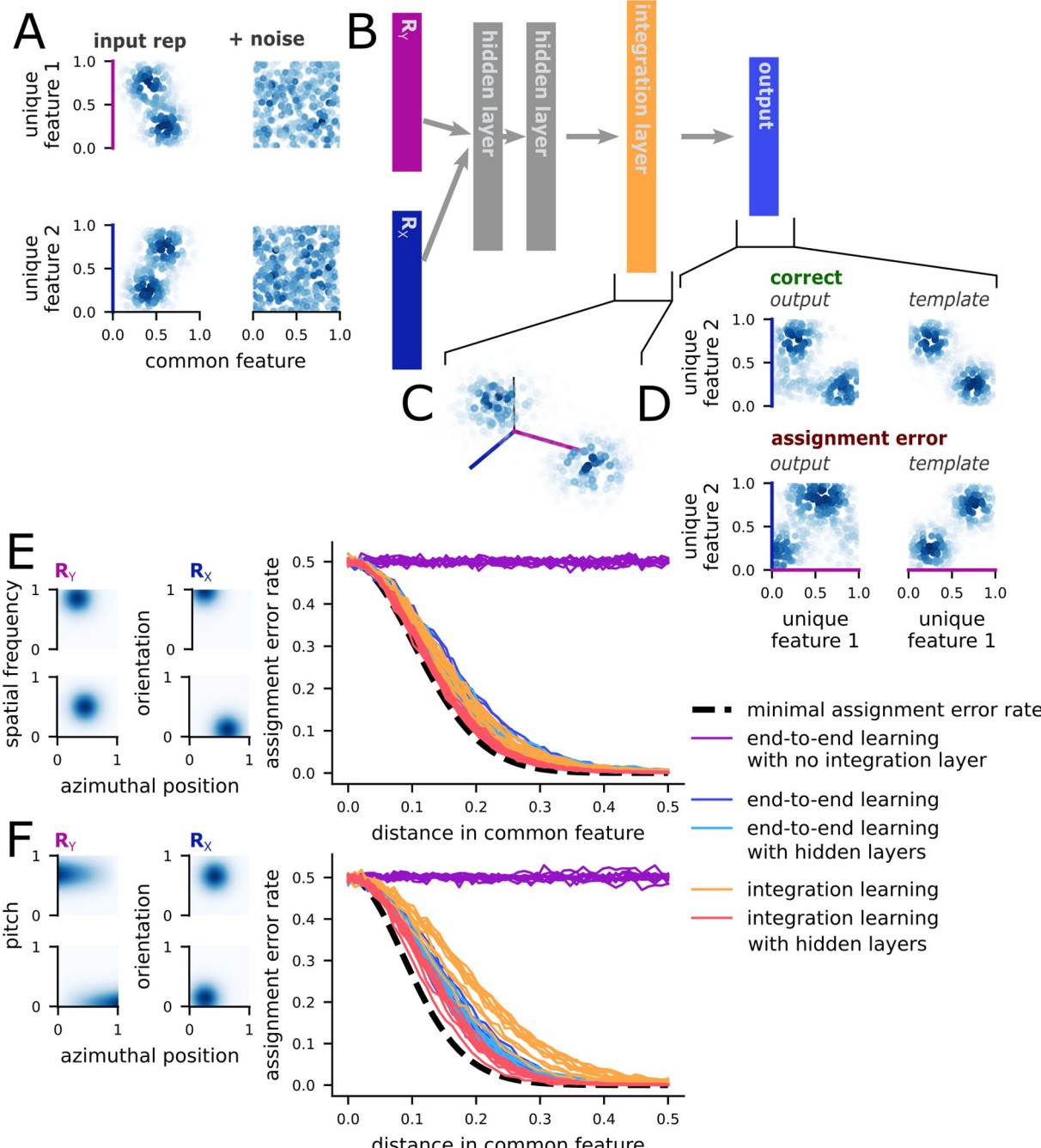

**Fig 5. The assignment problem can be solved with one feedforward layer. A** Example of the input, first without (left) and then with (right) additive noise. The visualization shows the activity level of units in the random RF population. Each point is one unit, and the point is positioned at the unit's RF location. The color shows the level of activity. The two inputs share one common feature (x-axis, $C = 1$) and each have one unique feature (y-axis). **B** Schematic of the feedforward integration, then marginalization network. In the full network, the input layers are followed by two hidden layers, then an integration layer, then the output layer. The output layer constructs a representation of only the two originally unique features of the stimuli, marginalizing out the common feature. **C** Example of target activity in the integration layer. The three features of the stimuli are represented together. The visualization is the same as **A**. **D** Example of a correct output (top) and an output with an assignment error (bottom). The output produced by an example network is shown on the left; the right shows the target output (top) or the prototypical assignment error (bottom). **E** (left) Both $R_X$ and $R_Y$ use the random Gaussian receptive field input developed in the previous sections. (right) The assignment error rates for stimuli with different distances in the common feature, using several variations on the network described in **B**. The end-to-end learning models are trained to reconstruct the marginalized representation from the inputs, either with or without intervening hidden and integration layers. The integration learning models are trained to construct both the integrated representation and the output target, with or without intervening hidden layers. **F** The same as **E** except $R_Y$ uses an auditory-like input format, where the common feature (putatively, position) is encoded through a linear ramp.

require extensive investigation of the representations in the integration layer that are outside of the scope of this paper, this result emphasizes the importance of the integration step. Further, it demonstrates that random RF codes can be used to solve the assignment problem in an optimally reliable way and that they outperform representations learned through backpropagation.

Finally, while assignment between two representations that both use Gaussian receptive fields is analogous to assignment between parallel regions in visual cortex, there is evidence that neurons in the auditory system primarily represent spatial position using ramp-like rather than Gaussian-like tuning functions [39–42] (and see Methods in Random ramp codes). While our theory does not depend on the assumption of a specific representational format, it is possible that solving the assignment problem may be more difficult for a simple neural network when the formats of the involved representations are different. To test this, we studied the case where $R_X$ had the same receptive field tuning as before, while $R_Y$ was composed of neurons that had a linear response to the common feature and Gaussian tuning to the unique feature (Fig 5f, left $R_X$). This mirrors that ramp-like position and restricted pitch tuning observed in auditory cortex. Despite this difference in format between the two inputs, the results were qualitatively similar to before, with the best networks nearly saturating the bound on performance provided by our theory (Fig 5f, right). However, average performance was further from the bound in this case, indicating that the difference in format does increase the difficulty of the task.

## 3 Discussion

We have described a neural instantiation of the assignment problem, which must be solved when the brain integrates distributed representations of multiple stimuli. We showed that assignment errors depend on the distance between stimuli in the shared representation space, and that increasing the dimensionality of that shared space will—on average—decrease their probability. However, this shared space comes at the cost of redundancy between the distributed representations. To make this cost more concrete, we studied assignment errors in random receptive field codes. In these codes, we showed an advantage for modular representations of high-dimensional stimuli in some conditions, which is consistent with modularity observed in cortex [29–36] (e.g., in the primate ventral and dorsal visual streams). Further, we showed that a simple neural network can reliably (and optimally) solve the assignment problem.

This work extends previous work on the assignment problem in several ways. First, previous studies have considered either a single stimulus in both modalities (one auditory and one visual stimulus in [21]) or a single stimulus that must be assigned to one stimulus from a larger array (using a cue to select a single stimulus from a remembered array in [16, 37]). Here, we have shown how the rate of assignment errors scales as more stimuli need to be integrated. Second, previous work used a single common feature for binding [16, 21, 37], while we have shown how assignment error rates scale for additional common features. Third, the formulation of the neural code that we use is similar to that used in [16]; however, we have derived novel closed form expressions for the total, local, and threshold error rates of these codes—using these expressions we investigate a wider variety of different neural architectures (e.g., splitting representations across different regions) than have been considered in previous work. Finally, we believe that our neural network approach to producing assigned representations is novel, as previous work only characterized the error rates that would be expected from an ideal observer [16, 21, 37]. Thus, we believe that this work contributes to our understanding of the assignment problem, building on top of this foundational work.

Our work produces several predictions for experimental data. First, efficient coding dictates that the redundancy in representations of the sensory world should be reduced as much as possible [53, 54]. However, our work shows that redundancy between the stimulus representations present in different brain regions is necessary to disambiguating multiple stimuli—other potentialy benefits of redundancy across and within brain regions are robustness to noise [50], damage, and interference [55]. As a result, we predict that brain regions that commonly represent multiple stimuli (e.g., sensory regions) will all have some redundancy with other brain regions. In some cases, the redundant information between two regions may be clear due to experimental design. However, in cases where redundant features are not readily identifiable, population analyses can be used to discover redundant information directly by building models that explain the activity in one region in terms of the activity in the other. Then the activity explained by this approach can be modeled in terms of known experimental variables. For instance, in this way, common features across the parallel ventral and dorsal visual streams in the primate visual system can be discovered (though some features, such as spatial position, have already been shown to be redundant across the two streams [51]).

Second, our work shows that explicit bound representations of the integrated and common features may be a necessary step in solving the assignment problem. Following previous work [16, 37, 38] and our neural network results, we believe that these bound representations will be instantiated through a conjunctive population geometry for the bound features—similar to the multi-dimensional receptive fields used above (though the representation need not have exactly this form). Thus, for a multi-sensory classification task, we expect to find these bound representations in multi-sensory association areas (such as posterior parietal cortex [56, 57], ventrolateral prefrontal cortex [58], and polysensory areas in the superior temporal sulcus [59]; also see [60]). For complex visual decision-making tasks where integration across the dorsal and ventral visual streams is required, we expect to find bound representations in brain regions receiving input from both streams (such as prefrontal and posterior parietal cortex [61, 62]) as well as regions within each stream that receive information from the other stream. For instance, recent work has shown combined representations of visual form and motion in the inferotemporal cortex [63], though their geometry was not characterized (i.e., it is unknown whether the representations are bound). Similar integrated representation of visual form and motion have also been found in the superior temporal polysensory area [64] (and in the middle [65] and body-selective patches within [66] the superior temporal sulcus).

In general, we expect that these intermediate representations will emerge prior to a pure representation of the decision variable—both in brain regions that are earlier in an established processing hierarchy as well as temporally earlier within brain regions that ultimately express representations of the decision variable. While the results from our neural network study underline the importance of this bound representation for computing the decision variable (Fig 5), a single neural population could simultaneously represent these distinct forms of information in separate subspaces of activity. These bound representations have also been identified as important for representation recoding [20]. Future work could identify whether these recoding and assignment operations can be performed simultaneously in the same integrated representation.

Third, our framework also makes predictions that can be tested with population recordings from uni-sensory brain regions, during the performance of a multi-sensory task (or, at least, a task that is believed to involve assignment). Our work predicts that behavioral assignment errors—i.e., a swap error [15–17], described above—will be correlated with errors in the representation of the common features specifically in the direction of other stimuli. For instance, in a visual-auditory integration task with multiple stimuli, we expect that trials in which a position decoder for one of the stimuli makes errors toward another stimulus will also be

associated with a greater probability of behavioral assignment errors. We do not expect errors in the representation of non-common features to have this same correlation with assignment errors—instead, we expect that they will be associated with local variance in the animal's report and would be detectable in continuous report tasks, such as [67].

Fourth, our work makes several predictions for what kinds of assignment errors will be more or less likely, many of which already have behavioral support in the time-assignment context from working memory experiments in humans. In particular, our framework predicts that the assignment error rate will increase for stimuli that are nearby in the common feature space, which is validated in [16, 17] (and see the related theoretical work in [16]). It also predicts that assignment error rates will decrease with the inclusion of additional features in the common feature space, which is validated here [18] (but see [19]). Finally, our framework predicts that the assignment error rate will increase with the number of stimuli, which is validated here [15, 16]. While this support is qualitative rather than quantitative, we believe that it lends strength to our predictions for neural data. Further, we believe that a strength of our framework is that it unifies several different types of assignment, including integration across distinct time points—as in the behavioral work here—and integration across distinct brain regions. However, these forms of integration are likely to differ to some degree in practice, even if only due to the kinds of redundant information available. Future work will be necessary to characterize what these potential differences mean for patterns of errors in different assignment contexts.

The simultaneous representation of multiple stimuli in neural activity is not fully understood. In particular, while we have provided a solution to the assignment problem, which must be solved when integrating representations of multiple stimuli that are distributed across different brain regions, distinct neural population subspaces within a single brain region, or even time points. However, the assignment problem is only one of several difficulties that arise when multiple stimuli are represented simultaneously. One additional problem is the segmentation, or clustering, of sensory information into a discrete set of causes (i.e., stimuli or objects). While there is extensive work on this process in psychology [1–3] and in machine learning [4, 5], the neural mechanisms are not fully understood (but see [6, 68]). A second additional problem that we have already mentioned is the representation of bound stimulus features in a single population of neurons; in this work, we have assumed that this representational problem has already been solved, perhaps by the conjunctive feature representations posited by other work [16, 37, 38] (but see [69–72] for alternatives). In the real brain, each of these problems is likely solved at many different stages of sensory processing (or even simultaneously), where lower-level components are clustered into higher-level features (e.g., combining points into an edge), a correct binding of those features is inferred (e.g., that edge is combined with a representation of its motion), then that bound set of features is represented (and can become a new, higher-level feature itself—e.g., perhaps the edge is part of a running dog). While we have focused on one part of this process, future work can integrate these parts and more fully explore how they interact—both with each other, and with other concerns (such as how multiple representations in a single neural population interact with each other, see [73]). This unifying work will be necessary to develop a full understanding of the rich ways that simultaneously representing multiple stimuli constrains the neural code.

We have also discussed the case of balanced assignment between two brain regions, where each region represents the same number of stimuli and there is a correct one-to-one mapping between those stimuli. However, this is a simplification: in many cases, there may be more representations in one region than the other (e.g., speakers who are not visible). Or, even when there are the same number of representations, a one-to-one mapping may not be correct (e.g., a speaker behind the subject and another person in view who is not speaking). This latter

situation has been studied for a single auditory and single visual stimulus [21, 74, 75]. This work found evidence that human observers infer a single cause for the two percepts only when they are nearby in space; otherwise, they infer independent causes—and further shows that this process is well-modeled by Bayesian inference, depending on the local variance of auditory and visual position estimates as well as the prior rate of common relative to distinct causes [21]. Future work can combine our approach to the assignment problem with this inference of the number of distinct causes in the environment. This could be achieved by considering more potential mappings between the sets of representations, along with a prior giving the strength of the expectation that each representation will be integrated. This generalization of our framework would lead to additional predictions for experimental data with unbalanced numbers of stimuli across distinct sensory modalities.

In summary, we have developed a general framework for understanding how the brain makes sense of distributed representations of multiple objects—that is, how the brain solves the representation assignment problem. This framework also points to several new directions in the study of neural codes: directions that explicitly contextualize neural representations in a distributed network of brain regions as well as move toward understanding how neural codes can be made suitable for encoding heterogeneous, multi-object environments.

## 4 Methods

### 4.1 Definition of the objects

An object set $S$ of size $N$ is $N$ independent samples from the multidimensional probability distribution $p(s)$—thus, $p(S) = \prod_i^N p(s_i)$. Each $s_i$ is a vector of length $K$, so $p(s)$ is a probability distribution over a $K$-dimensional space. Each of these dimensions represents a feature of the object, such as color, spatial frequency, pitch, or orientation. In the majority of the paper, we assume that the objects are uniformly distributed in the space—that is, each point in the $K$-dimensional volume is equally likely to occur.

### 4.2 Definition of the representations

We focus on the representation of our $K$-dimensional objects in two brain regions, $R_X$ and $R_Y$. We assume that both regions have some common and some unique information. That is, neither region encodes all $K$ object features. This is guaranteed to be the case when $R_X$ and $R_Y$ represent early sensory areas from different sensory modalities—and there is evidence that a balance of unique and common information is preserved across hierarchies of sensory brain regions even within single modalities, such as in the primate visual system [29–32].

Formally, $R_X$ encodes a subset of the $K$ total object features, denoted as $F_X$—and similarly for $R_Y$. Each of the features can be identified by their index from 1 to $K$, and region $R_X$ is said to encode feature $i \in \{1, 2, \ldots, K\}$ with local distortion $D_X^i$, which is the variance of an optimal estimator for the value of feature $i$ from the neural activity in region $R_X$—and, again, similarly for $R_Y$.

Thus, the subset of features represented in $R_X$ that are not represented in $R_Y$ are the unique information from $R_X$—that is, $F_X \setminus F_Y$. When $R_X$ is an auditory region and $R_Y$ is a visual region, then these unique features might include representations of pitch and timbre. Further, the subset of features represented in both $R_X$ and $R_Y$ are the common information, which is essential for solutions to the assignment problem. The size of this intersection $|F_X \cap F_Y| = C_{XY} = C$ has important consequences for the assignment error rate, and the achievable local distortion when the representation capacity is constrained.

Here, we study the reliability of inferences about the original set of $N$ objects $S$, each of which are described by $K$ feature dimensions, from the neural activity in two distinct regions, $R_X$ and $R_Y$, which both encode some common and some unique information about the objects. Inferring the original object set from two distinct representations requires the two sets of representations to be combined with each other, which we refer to as assignment.

### 4.3 Definition of assignment errors

As mentioned above, to make inferences about the whole object set from two distinct sources of information (i.e., $R_X$ and $R_Y$), the two sets of representations must be integrated with each other. When there is only one object ($N = 1$), then this integration is trivial as there is only one possible one-to-one mapping between the two sets of representations, and this mapping is correct. However, when there is more than one object ($N > 1$), then assignment errors become possible. In particular, for $N$ objects, there are $N!$ possible one-to-one mappings (i.e., assignments) between the two sets of representations, $\hat{X}$ and $\hat{Y}$. Thus, if there is no information about which mapping to select, assignment errors have the probability $1 - 1/N!$, which is near 1 for even relatively small $N$.

Formally, we can frame the mapping selection problem as an inference about which of the possible maps $M$ is most likely to account for the observed representations $\hat{X}$ and $\hat{Y}$ in $R_X$ and $R_Y$, respectively. This can be written as,

$$\begin{aligned} p(M|\hat{X}, \hat{Y}) \quad &= \frac{p(\hat{X}, \hat{Y}|M)p(M)}{p(\hat{X}, \hat{Y})} \\ &\propto p(\hat{X}|\hat{Y}, M)p(\hat{Y}|M) \\ &\propto p(\hat{X}|\hat{Y}, M) \end{aligned} \tag{M1}$$

where we proceed by first assuming that the prior probability of each map is the same (i.e., $p(M)$ is uniform) and that the representations in one region do not depend on the map (i.e., $p(\hat{Y}|M) = p(\hat{Y})$, and similarly for $\hat{X}$). Thus, we are left with a single term, $p(\hat{X}|\hat{Y}, M)$, that gives the likelihood of the representations observed in one region (here, $\hat{X}$, but $X$ and $Y$ are interchangeable) conditioned on a particular map $M$ and the representations observed in the other region ($\hat{Y}$). If $\hat{X}$ and $\hat{Y}$ are independent of each other, then $p(\hat{X}|\hat{Y}, M) = p(\hat{X})$, as above, and all maps are equally likely. So, it is dependence between the representations in $R_X$ and $R_Y$ that enables the correct assignment to be selected at a rate better than chance. Thus, dependence between those representations is necessary for a reliable solution to the assignment problem. In general, this observation already indicates to us that we should expect pairs of brain regions to encode some common information—so that the assignment problem can be solved—and some unique information—due to both distinct sensory systems, but also due to considerations related to efficient coding, which we make explicit below.

**4.3.1 Probability of assignment errors.** Given the above inference process, we can now characterize the likelihood that an assignment error occurs given different levels of common information shared between $\hat{X}$ and $\hat{Y}$. From our formalization of the representations above, we know that each object feature $i$ is estimated from the activity of neurons in region $R_X$ with variance $D_X^i$, and similarly for $R_Y$. Here, we assume that these estimates are unbiased and Gaussian distributed, with mean equal to the true value of the object feature and variance as given. The Gaussian distribution is the maximum entropy distribution for fixed mean and variance, which means that these estimates will contain less information about the true value of

the object feature than any other distribution with the same mean and variance—thus, this assumption represents an upper bound on the difficulty of the integration task.

Using this formalization, we can write an explicit form for Eq M1,

$$
\begin{aligned}
p(M|\hat{X}, \hat{Y}) &\propto \prod_i^N \prod_j^C \exp\left(-\frac{(\hat{x}_i^j - M_i\hat{y}^j)^2}{2(D_X + D_Y)}\right) \\
\log(p(M|\hat{X}, \hat{Y})) &\propto -\sum_i^N \sum_j^C (\hat{x}_i^j - M_i\hat{y}^j)^2
\end{aligned}
\tag{M2}
$$

where $\hat{x}_i^j$ and $\hat{y}_i^j$ are the values of feature $j$ for object $i$ from the $N \times C$ estimation matrix of the $C$ commonly represented features for each of the $N$ objects. The common features are ordered consistently across $\hat{x}$ and $\hat{y}$, while the $N$ different objects are not. The map $M$ is an $N \times N$ permutation matrix. Thus, in finding the most likely map, we would search over the $\binom{N}{2}$ possible permutation matrices to find the one that maximizes Eq M2. This is equivalent to finding the permutation matrix that minimizes the sum squared distance between integrated object representation pairs in the $C$-dimensional shared feature space.

From the above, it follows that an assignment error occurs precisely when the representation of two objects cross over each other in one of the two brain regions, but not in both—as schematized in Fig 1C, red arrow and assignment lines. For the commonly represented feature values of two objects, $x_1^c$ and $x_2^c$, the probability that this crossover happens in $R_X$ depends on the distribution of the distances between their estimates,

$$
\hat{x}_2^c - \hat{x}_1^c \sim \mathcal{N}(\delta, 2D_X)
$$

where $\delta$ is the distance between the true values of $x_1^c$ and $x_2^c$ (i.e., $\delta = x_2^c - x_1^c$) and we assume, without loss of generality, that $x_1^c < x_2^c$. Since $\delta$ is still Gaussian distributed, we write the probability that the estimate of the first object becomes greater than the estimate of the second object (i.e., that $\hat{x}_1^c > \hat{x}_2^c$) as

$$
P(\text{cross in } R_X) = Q\left(\frac{-\delta}{\sqrt{2D_X}}\right)
$$

where $Q(.)$ is the cumulative distribution function for the standard Gaussian distribution.

Following this, the full probability of an assignment error incorporates the probability that the cross occurs in $R_X$ or $R_Y$ as well as that it occurs in both (which would not result in an assignment error). We write this probability as,

$$
F(\delta) = Q\left(\frac{-\delta}{\sqrt{2D_X}}\right) + Q\left(\frac{-\delta}{\sqrt{2D_Y}}\right) - Q\left(\frac{-\delta}{\sqrt{2D_X}}\right)Q\left(\frac{-\delta}{\sqrt{2D_Y}}\right)
$$

where the final term is the probability that a cross occurs in both regions. While we have discussed a single common feature here, this expression is general, and applies for any value of $C$, so long as the local distortion $D_X$ and $D_Y$ for all of the common features is the same within each region, which we assume in the majority of the text. This expression already gives us insight into how assignment errors depend on $D_X$ and $D_Y$ for stimuli at some distance $\delta$ in a common feature space. However, in general, assignment errors also depend on how likely it is that two stimuli at a particular distance will be observed—that is, on $p_C(\delta)$. In the main text, we develop this dependence, as well as a dependence on the number of objects, which results in Eq 2.

**4.3.2 Assignment error rate approximation for $C = 1$.** For one overlapping object feature ($C = 1$), we derive an approximate closed form for Eq 2. The derivation is as follows,

$$
\begin{aligned}
AE_1 \quad &\leq \binom{N}{2} \int_0^s d\delta \, p_C(\delta) \, F(\delta) \\
&= \binom{N}{2} \int_0^s d\delta \, \frac{2(s-\delta)}{s^2} \left( Q\left(\frac{-\delta}{\sqrt{2D_X}}\right) + Q\left(\frac{-\delta}{\sqrt{2D_Y}}\right) - Q\left(\frac{-\delta}{\sqrt{2D_X}}\right) Q\left(\frac{-\delta}{\sqrt{2D_Y}}\right) \right) \\
&\approx \binom{N}{2} \int_0^s d\delta \, \frac{2(s-\delta)}{s^2} \left( Q\left(\frac{-\delta}{\sqrt{2D_X}}\right) + Q\left(\frac{-\delta}{\sqrt{2D_Y}}\right) \right) \\
&= \binom{N}{2} \frac{2}{s^2} \int_0^s d\delta \, (s-\delta) Q\left(\frac{-\delta}{\sqrt{2D_X}}\right) + (s-\delta) Q\left(\frac{-\delta}{\sqrt{2D_Y}}\right)
\end{aligned}
$$

since the two terms in this sum are analogous to each other, we deal with them separately before combining.

So, for the first part of each term,

$$
\begin{aligned}
\int_0^s d\delta \, s \, Q\left(\frac{-\delta}{\sqrt{2D_i}}\right) &= -s\sqrt{2D_i} \left( -\frac{\delta}{\sqrt{2D_i}} Q\left(-\frac{\delta}{\sqrt{2D_X}}\right) + \phi\left(-\frac{\delta}{\sqrt{2D_X}}\right) \right) \bigg|_0^s \\
&= -s\sqrt{2D_i} \left( \phi\left(-\frac{s}{\sqrt{2D_i}}\right) - \frac{s}{\sqrt{2D_i}} Q\left(-\frac{s}{\sqrt{2D_i}}\right) \right) + s\sqrt{\frac{D_i}{\pi}} \\
&\approx s\sqrt{\frac{D_i}{\pi}}
\end{aligned}
$$

where $\phi$ is the standard normal density function and the approximation in the last line holds when $s \gg D_i$, which is the regime we focus on for the main text.

Now, for the second part of each term,

$$
\begin{aligned}
-\int_0^s d\delta \, \delta \, Q\left(\frac{-\delta}{\sqrt{2D_X}}\right) &= -4D_i \left( \left(\frac{\delta^2}{2D_i} - 1\right) Q\left(-\frac{\delta}{\sqrt{2D_i}}\right) - \frac{\delta}{\sqrt{2D_i}} \phi\left(-\frac{\delta}{\sqrt{2D_i}}\right) \right) \bigg|_0^s \\
&= -4D_i \left( \left(\frac{s^2}{2D_i} - 1\right) Q\left(-\frac{s}{\sqrt{2D_i}}\right) - \frac{s}{\sqrt{2D_i}} \phi\left(-\frac{s}{\sqrt{2D_i}}\right) \right) - 2D_i \\
&\approx -2D_i
\end{aligned}
$$

where, again, the approximation in the last line holds when $s \gg D_i$.

Then, combining the two expressions above,

$$
\frac{2}{s^2} \int_0^s d\delta \, (s-\delta) \, Q\left(-\frac{\delta}{\sqrt{2D_i}}\right) \approx \frac{2}{s}\sqrt{\frac{D_i}{\pi}} - \frac{4}{s^2} D_i
$$

Before, finally, combining the terms corresponding to $D_X$ and $D_Y$ gives,

$$
\begin{aligned}
AE_1 &\approx \binom{N}{2}\left[\frac{2\sqrt{D_X}}{s\sqrt{\pi}} - \frac{4D_X}{s^2} + \frac{2\sqrt{D_Y}}{s\sqrt{\pi}} - \frac{4D_Y}{s^2}\right] \\
&= \binom{N}{2}\left[\frac{2\sqrt{D_X + D_Y}}{s\sqrt{\pi}} - \frac{4(D_X + D_Y)}{s^2}\right] \\
&\approx \binom{N}{2}\frac{2\sqrt{D_X + D_Y}}{s\sqrt{\pi}}
\end{aligned}
$$

This final form is given as Eq 3 in the main text.

**4.3.3 Assignment error simulations.** We compare the theory developed above to simulations from a generative process that follows many of the assumptions made above. In particular, we sample $N$ stimuli as described above. Then, we add normally distributed noise, with variance $D_X$ and $D_Y$, to the features represented in each of the two regions (including common features). Then, we build a distance matrix from the pairwise distances in the common feature space between all stimuli. Finally, we solve the balanced assignment problem for this distance matrix using a standard algorithm (implemented in scipy [76]). A sample is counted as an error if there is at least one incorrect assignment.

## 4.4 Random Gaussian receptive field codes

We consider codes for a uniformly distributed $K$-dimensional stimulus. Without loss of generality, we take the features of the stimulus to be between 0 and 1. The code consists of $N$ units with Gaussian receptive fields. We assume that all units have the same field widths $w$ and height $P$, but random centers $\mu^i$. In particular, the response of a neuron $i$ is given by

$$
r_i(x) = P\exp\left[-\sum_j^K \frac{(x_j - \mu_j^i)^2}{2w_j^2}\right] + v
$$

where $v \sim \mathcal{N}(0, \sigma_N)$ and, here, we assume that $w = w_j \forall j$.

We model the response to multiple stimuli as the sum of responses to each individual stimulus, so that

$$
R_i(X) = \sum_{x \in X} r_i(x) + v
$$

where $X$ is a set of stimuli. Note that the noise $v$ is only added once.

**4.4.1 The spiking energy of the code.** First, we compute the average squared $L_2$-norm of the code response across all stimuli, which we will refer to as $V$ and which we use as a measure of the spiking metabolic energy used by the code,

$$
\begin{aligned}
V &= \mathbb{E}_x\left[\sum_i^N r_i^2(x)\right] = \sum_i^N \int dx_1 \ldots \int_0^1 dx_K p(x) r_i(x)^2 \\
&= \sum_i^N \int_0^1 dx_1 \ldots \int_0^1 dx_K p(x) P\exp\left[-\sum_j^K \frac{(x_j - \mu_j)^2}{w^2}\right] \\
&= P^2 \sum_i^N \prod_j^K \int_0^1 dy_j p(y_j) \exp\left[-\frac{y_j^2}{w}\right] \\
&= P^2 \sum_i^N \prod_j^K \int_0^1 dy_j (2 - 2y_j) \exp\left[-\frac{y_j^2}{w}\right]
\end{aligned}
\tag{M3}
$$

where Eq M3 changes $y_j^2 = (x_j - \mu_j)^2$. Because both $x_j$ and $\mu_j$ are uniformly distributed random variables, we know that $|x_j - \mu_j| = y_j \sim T(0, 1, 0)$ where $T(a, b, c)$ is a triangular distribution.

We then split the above into two terms. For the first,

$$\int_0^1 dx_j 2 \exp\left[-\frac{y_i^2}{w^2}\right] = \sqrt{\pi} w \,\mathrm{erf}\left(\frac{y}{b}\right)\Big|_0^1$$
$$= \sqrt{\pi} w \,\mathrm{erf}\left(\frac{1}{w}\right)$$

and the second,

$$\int_0^1 dx_j 2 y_i \exp\left[-\frac{y_i^2}{w^2}\right] = -w^2 \exp\left[-\frac{x^2}{w^2}\right]\Big|_0^1$$
$$= w^2\left(1 - \exp\left[-\frac{1}{w^2}\right]\right)$$

So, together,

$$V = P^2 \sum_i^N \prod_j^K \sqrt{\pi} w \,\mathrm{erf}\left(\frac{1}{w}\right) - w^2\left(1 - \exp\left[-\frac{1}{w^2}\right]\right)$$
$$= NP^2 \prod_j^K \sqrt{\pi} w \,\mathrm{erf}\left(\frac{1}{w}\right) - w^2\left(1 - \exp\left[-\frac{1}{w^2}\right]\right)$$
$$= NP^2\left[\sqrt{\pi} w \,\mathrm{erf}\left(\frac{1}{w}\right) - w^2\left(1 - \exp\left[-\frac{1}{w^2}\right]\right)\right]^K$$

and, further, we will tend to deal with $w < .5$, so,

$$V \approx NP^2\left[\sqrt{\pi} w - w^2\right]^K$$

and

$$P = \sqrt{\frac{V}{N[\sqrt{\pi} w - w^2]^K}}$$

## 4.5 The Fisher information of the code

We will use the Fisher information as a measure of the magnitude of local errors via the Cramer-Rao bound. In our framework, the Fisher information along a particular dimension $j$ is given by

$$I_j(x) = \mathbb{E}_r\left[\left(\frac{\partial}{\partial x_j} \log p(r|x)\right)^2\right]$$
$$= \mathbb{E}_r\left[\left(-\frac{\partial}{\partial x_j} \frac{(r - \bar{r}(x))^2}{2\sigma_N^2}\right)^2\right]$$

We want to use the Fisher information to understand the average code performance across the whole stimulus space, rather than only for single points in the stimulus space. There is a complication here: the Fisher information is related to the MSE of an optimal, unbiased

estimator by the Cramer-Rao bound, which states:

$$\mathrm{MSE}(x_j) \geq \frac{1}{I_j(x)}$$

so, we would really like to evaluate

$$\mathbb{E}_x[\mathrm{MSE}(x_j)] \geq \mathbb{E}_x\left[\frac{1}{I_j(x)}\right]$$

However, we were not able to evaluate this analytically. Instead, we evaluate

$$\mathbb{E}_x[\mathrm{MSE}(x_j)] \geq \frac{1}{\mathbb{E}_x[I_j(x)]}$$

which provides a good approximation of the former quantity so long as the variance of $I_j(x)$ over $x$ is small relative to the mean.

So,

$$
\begin{aligned}
I_j &= \mathbb{E}_x[I_j(x)] = \frac{P}{\sigma_N^2}\sum_i^N\prod_k^K\int_0^1 dy_k p(y_k)\exp\left[-\frac{y_k^2}{w^2}\right]\int_0^1 dy_j p(y_j)\frac{y_j^2}{w^2}\exp\left[-\frac{y_j^2}{w^2}\right] \\
&= N\frac{P^2}{\sigma_N^2 w^4}\left[\sqrt{\pi}w\mathrm{erf}\left(\frac{1}{w}\right) - w^2\left(1 - \exp\left[-\frac{1}{w^2}\right]\right)\right]^{K-1} \\
&\quad \left[\frac{1}{2}w^2\left(\sqrt{\pi}w\mathrm{erf}\left(\frac{1}{w}\right) - 2\exp\left[-\frac{1}{w^2}\right]\right) - w^2\left(w^2 - (w^2+1)\exp\left[-\frac{1}{w^2}\right]\right)\right]
\end{aligned}
$$

following a similar sequence as above for $V$. For small relatively small $w$ (roughly, $w < .5$), we can approximate this expression as

$$
\begin{aligned}
I &\approx \frac{NP^2}{\sigma_N^2 w^4}\left[\sqrt{\pi}w - w^2\right]^{K-1}\left[\frac{1}{2}w^3\sqrt{\pi} - w^4\right] \\
&= \frac{NP^2}{\sigma_N^2}w^{K-2}\left[\sqrt{\pi} - w\right]^{K-1}\left[\frac{1}{2}\sqrt{\pi} - w\right] \\
&= \frac{V}{\sigma_N^2}\frac{w^{K-2}}{\left[\sqrt{\pi}w - w^2\right]^K}\left[\sqrt{\pi} - w\right]^{K-1}\left[\frac{1}{2}\sqrt{\pi} - w\right] \\
&= \frac{V}{\sigma_N^2 w^2}\frac{\left[\sqrt{\pi} - w\right]^{K-1}}{\left[\sqrt{\pi} - w\right]^K}\left[\frac{1}{2}\sqrt{\pi} - w\right] \\
&= \frac{V}{\sigma_N^2 w^2}\frac{\frac{1}{2}\sqrt{\pi} - w}{\sqrt{\pi} - w}
\end{aligned}
$$

**4.5.1 The threshold error rate of the code.** We compute the rate of threshold errors by following the derivations from [43, 50]. In particular, threshold errors occur when the response in the neural population is closer to a non-stimulus representation than the current stimulus representation. To proceed, we discretize the population response, such that there are $1/S(w, K)$ distinct subpopulations, only one of which is active for a given stimulus, where $S(w, K)$ is the volume of a $K$-dimensional sphere with radius $2w$. Importantly, this ratio increases both as $K$ increases and $w$ decreases. Since we know the $L_2$-norm of the code is $\sqrt{V}$, we know

that the distance between the representation of the correct stimulus and the representation of a random, distant stimulus is $\sqrt{2V}$. So, we need to find the probability that the noise crosses half that distance, to become closer to the representation of the distant stimulus. This is given by

$$
\begin{aligned}
p_{\text{switch}} \quad &= Q\left(-\frac{\sqrt{2V}}{2\sigma_N}\right) \\
&= \frac{1}{2}\text{erfc}\left(-\frac{\sqrt{V}}{2\sigma_N}\right) \\
&\approx \frac{1}{2}\exp\left[-\frac{V}{4\sigma_N^2}\right]\frac{2\sigma_N}{\sqrt{V\pi}} \\
&= \frac{\sigma_N}{\sqrt{V\pi}}\exp\left[-\frac{V}{4\sigma_N^2}\right]
\end{aligned}
$$

where the third line follows from an approximation of the error function for large argument.

Then, by a union bound, we find that,

$$
p_{\text{thr}} \approx \frac{1}{S(w,K)}\frac{\sigma_N}{\sqrt{V\pi}}\exp\left[-\frac{V}{4\sigma_N^2}\right]
$$

However, we find in simulations that this underestimates the probability of threshold errors. The reason is simple: $\sqrt{V}$ represents the average $L_2$-norm. However, in the random RF code, there is significant variance around this average for different stimuli. As a consequence, stimuli with low $L_2$-norms will be exponentially more likely to have threshold errors, but this is not accounted for by our equation. To incorporate this, we replace $V$ with $V_\lambda$, where,

$$
V_\lambda = \max(V - \lambda\sigma_V(w), 0)
$$

where $\sigma_\lambda(w)$ is the standard deviation of $V$ across stimuli for a random RF code, and it depends on $w$. This quantity is approximated analytically in the supplement (see S1 Text).

Given this correction and setting $\lambda = 2$, we find good agreement between the resulting total predicted MSE and the empircal MSE (Fig 3D and 3E).

**4.5.2 Total MSE of the code.** Finally, we can write the total approximate MSE for small $w$ as follows,

$$
\text{MSE} \approx \frac{2w^2\sigma_N^2}{V} + p_{\text{thr}}(w, V, \sigma_V)\frac{1}{6}
$$

where the $\frac{1}{6}$ is the average squared size of a threshold error, calculated from the triangle distribution. We find that this expression has good agreement with the MSE estimated from simulations (Fig 3D and 3E).

**4.5.3 RF code simulations.** To generate the simulation traces for the RF codes, we sampled stimuli from a uniform distribution over $K$ dimensions, then obtained the representations of those stimuli in a particular instantiations of random RF codes (depending on the number of regions in the code). Then, within each code, we used maximum likelihood decoding to obtain an estimated stimulus and compared that estimate with the true stimulus.

## 4.6 Random ramp codes

To explore a situation more similar to representations in the mammalian auditory system, we also use neural representations with the following form:

$$r_i(x) = 2P\left(x_R - \frac{1}{2}\right) \cdot u_R \exp\left[-\sum_j^{K_G} \frac{(x_{G,j} - \mu_j^i)^2}{2w_j^2}\right] + v$$

where the stimulus $x$ is now divided into dimensions that are encoded with ramp-like tuning $x_R$ and receptive field-like tuning $x_G$. The neuron is also assumed to have a preferred axis $u_R$ in the space of features encoded with ramp-like tuning. In our simulations, $u_R$ is chosen to be a random unit vector.

In the case of the auditory system, single neurons are thought to respond to a specific frequency band (i.e., a receptive field-like representation) while having the intensity of their response modulated by the position of the sound (i.e., a ramp-like representation) [39–42].

## 4.7 Calculating the redundancy between representations

We quantify the redundancy between $R_X$ and $R_Y$ as the mutual information between $\hat{X}$ and $\hat{Y}$,

$$
\begin{aligned}
R \quad &= I(X; Y) = H(X) - H(X|Y) \\
&= K_X \frac{1}{2} \log 2\pi s^2 - (K_X - C)\frac{1}{2} \log 2\pi s^2 - C\frac{1}{2} \log 2\pi (D_X + D_Y) \\
&= C\frac{1}{2} \log 2\pi s^2 - C\frac{1}{2} \log 2\pi (D_X + D_Y) \\
&= \frac{C}{2} \log \frac{s^2}{D_X + D_Y} \\
&= \frac{C}{2} \log \frac{s^2(1 - \Delta D^2)}{4D_S^2}
\end{aligned}
$$

where $R$ is the redundancy between the representations in region $X$ and $Y$ (in nats). As discussed throughout this work, this redundancy is crucial for solving the assignment problem. The redundancy is proportional to the number of commonly represented features $C$—so, increasing $C$ will produce relatively large increases in redundancy, and, as we have seen, can be expected to effectively reduce assignment errors. Further, increasing the asymmetry of feature representations $\Delta D$ reduces the level of redundancy between the $R_X$ and $R_Y$—so, as anticipated, increasing this asymmetry will increase the assignment error rate.

This redundancy represents the cost of our solution to the assignment problem. Thus, we desire a solution to the assignment problem that achieves a particular assignment error rate and local distortion magnitude while using as few bits—and, in particular, as few redundant bits—as possible.

## 4.8 The random RF integration model

The stimuli for the integration model are described by $K = 3$ dimensions. On each "trial," we sample two uniformly distributed, random stimuli. Two features are represented in each of the two input populations. The two populations each represent $C = 1$ overlapping and one unique feature, with $N = 400$ units. Each population is noisy and has SNR $= \sqrt{20}$.

When present, the integration layer has $N = 2000$ units. The training targets are generated by obtaining representations from a $K = 3$-dimensional random RF code. When present, the hidden layers are two layers of 500 units, with no specific training target. The output

population is $N = 400$ units, and the training target is given by samples from a random RF code for the two unique features in the input representation.

**4.8.1 End-to-end learning.**   The network is trained to produce the output population directly. The different network architectures (integration or no integration, hidden layers or no hidden layers) indicate how many and what kind of intervening layers are present; the training objective does not change.

**4.8.2 Integration learning.**   The network is trained to produce the activity from a random RF code in the integration layer (i.e., the full $K = 3$-dimensional representation). The output is learned as a second objective for backpropagation.

**4.8.3 Training procedure.**   The nonlinear layers are learned by backpropagation with stochastic gradient descent, using the Keras Tensorflow [77] environment and the Adam optimizer. The batch size is set to 200 and the network is trained for a maximum of 200 epochs on 50000 input samples with early stopping (if performance on a validation set stops improving for 2 epochs). In practice, the early stopping is often triggered after around 20 epochs.

## Supporting information

**S1 Text. Additional results and derivations, including two supplementary figures.** In the supplementary text, we include additional detail on several points discussed in the main text as well as explore several related topics. In particular, we

1. Consider the case of asymmetric feature representations.

2. Consider the case of nonlinear mappings between the two feature representations.

3. Derive an expression for the variance of spiking energy of the RF codes.
   (PDF)

## Acknowledgments

We are thankful to Stephanie Palmer, Ji Xia, and Matteo Alleman for their comments on an earlier version of this work. We are also thankful to Allison Ong, Mahham Fayyaz, and Stephanie Hoker for administrative support.

## Author Contributions

**Conceptualization:** W. Jeffrey Johnston, David J. Freedman.

**Formal analysis:** W. Jeffrey Johnston.

**Funding acquisition:** W. Jeffrey Johnston, David J. Freedman.

**Investigation:** W. Jeffrey Johnston.

**Methodology:** W. Jeffrey Johnston.

**Project administration:** David J. Freedman.

**Resources:** David J. Freedman.

**Supervision:** David J. Freedman.

**Visualization:** W. Jeffrey Johnston.

**Writing – original draft:** W. Jeffrey Johnston.

**Writing – review & editing:** W. Jeffrey Johnston, David J. Freedman.

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
