## [Decision Letter · Decision Letter 0]

27 Apr 2023

Dear Johnston,

Thank you very much for submitting your manuscript "Redundant representations are required to disambiguate simultaneously presented complex stimuli" for consideration at PLOS Computational Biology. As with all papers reviewed by the journal, your manuscript was reviewed by members of the editorial board and by several independent reviewers. The reviewers appreciated the attention to an important topic. Based on the reviews, we are likely to accept this manuscript for publication, providing that you modify the manuscript according to the review recommendations.

Sincerely,

Tianming Yang

Academic Editor

PLOS Computational Biology

Lyle Graham

Section Editor

PLOS Computational Biology

Reviewer's Responses to Questions

**Comments to the Authors:**

Reviewer #1: In this study, Johnston and Freedman proposed a theoretical solution for how distributed representations are integrated (the assignment problem): that one or more common stimulus features are represented in pairs of relevant brain regions. They further implemented the solution in a biological plausible receptive field code, and showed the tradeoff between assignment errors and redundancy. Lastly, they trained a feedforward neural network to show that the optimal solution can be learned when an integration layer was included.

Overall, I think this is a technically sound study that addresses an important question in the field, and has potential implications for experimental work in different domains. Meanwhile, I have a few clarification questions:

1. This work focuses on theoretical solutions of the assignment problem, but I think it would be greatly helpful if the authors could provide more concrete examples, or more concrete predictions, of the solution. For instance, it is reasonable to expect such integrated representations to be found in multi-sensory associative regions, in the example of visual-auditory integration; but how exactly would common features develop and integrate in the case of multi-region representations within a single sensory system? From how I read it, the authors were referring to regions in different sensory pathways (e.g., the dorsal and ventral visual pathways, as in ref. 30); however, the authors also talked about regions within a sensory pathway (e.g., along the ventral visual stream, as in ref. 49). So, making it more explicit, and possibly discussing where in the brain to expect such integrated representations, could help.

2. In Section 2.4, the authors discussed two errors that contribute to combined MSE: local and threshold errors. The tradeoff between the two errors is critical in determining the optimization of MSE. However, in the following Section 2.5, all analyses were restricted to RF codes with negligible threshold error rates, making the assignment problem a lot simplified. I did not find the rationale for doing this, and I would like to see more clarification and discussion on this issue.

3. Variants of the assignment problem have been studied in previous literature, such as the feature binding problem (e.g., Schneegans & Bays, 2017), and the causal inference problem (e.g., Kording et al., 2007). Although these alternative models were briefly introduced in Introduction, a more thorough discussion on the similarities and differences between the current and previous models would still be necessary. For example, the paper discussed successful integration and incorrect integration (i.e., swap error as in feature binding), but how would segregation (e.g., separate sources of visual and auditory information as in causal inference) occur and quantified in the current solution?

Reviewer #2: This manuscript describes a computational solution to the problem of assigning multiple properties (represented by distinct regions/modalities) to multiple objects, a problem related to causal inference in multisensory integration but with a slightly different twist. The approach seems elegant and creative, the results are interesting and the paper very well written. I think it should be published and will be of interest to many.

However, although logically and mathematically sound, as a theory of brain function I found it somewhat lacking. The premise as explained out of the gate seems a little farfetched and possibly circular. It assumes a kind of object-level organization that may not exist in most sensory cortices, at least not early/primary areas. Labeling the relevant visual area V1 in Fig. 1A (instead of, say, IT) for high-level representations like dog/cat is of course incorrect (not sure about A1), which doesn't make a good first impression. That’s a minor thing, but I think the issue is deeper one, and relates to two poorly defined concepts: ‘object’ and ‘region’ (see below). I also have a third general comment about the theory’s (in my view insufficient) contact with neurophysiology and anatomy.

1.

First, the conception of the assignment problem assumes that the segmentation and binding problems (within a modality) have already been solved, and all that remains is to assign features to the appropriate objects. I wrote a long diatribe about this before realizing that the paper covers it concisely in the Discussion — shame on me for not reading all the way through first, but it means that many readers will have this impression too, and you should confront this limitation earlier, i.e. in the Introduction. I still think it might weaken the premise due to circularity, since one could argue segmentation and binding already require a solution to the assignment problem. But I suppose that’s no reason to hold up publication; the community can be the judge.

2.

Second, the notion of a ‘brain region’ (especially critical for section 2.5), and its relationship to perception, is underdeveloped. The authors write: “The assignment problem must be solved when stimulus features are represented in distinct brain regions” — Why? Because perception requires convergence of information onto a single region (or single neurons)? That’s a big assumption. What region might that be? Isn’t our modern understanding that perception and behavior emerge from the collective activity of many interconnected populations/regions? If so, it becomes harder to motivate the problem, I think.

And even if ‘region’ is the right level of anatomical organization to think about here, how do you define the term? Neural populations are separated spatially but connected synaptically—do two highly interconnected but spatially disparate populations count as one region? What about two very nearby but only weakly connected populations? Or is a region defined by a complete representation of some stimulus dimension, like how boundaries between visual and somatosensory areas are defined by where the retinotopic or somatotopic map mirror-flips and begins to repeat? This works for most early sensory cortices but probably not later ones (e.g. where ‘dog’ and ‘cat’ live).

The point is not that this paper needs to solve these quasi-philosophical issues (although there are some philosophers out there who might be worth consulting). But it could be made clearer what is assumed about the 'read-out' of neural activity that necessitates the solution you propose. In other words, after the assignment problem is solved, what sort of neural representation are you left with, and what is the next step in the causal chain to behavior?

3.

Lastly, the theory could be better linked to the neurobiology, assuming that is a goal the authors seek. Aspects of this issue have been touched on above: defining regions better and being more up-front about the requirement for object-level representations (implicating later stages of processing). But when it comes to specific predictions for experiments, the relevant paragraph in Discussion (387-396), although a good start, comes up short.

For instance, “Brain regions that commonly represent multiple stimuli will all have some redundancy with other brain regions” — that’s a bit vague and probably trivially true (or already known to be true), therefore not very useful as a prediction.

And, “Our framework can be used to predict assignment error rates based on the level of redundancy between two brain regions” — how exactly would that play out? Can you be more specific?

What about RF codes? Can you say anything about (for example) how the width of RFs in real data compare to what is ‘optimal’ or predicted by the model? This would probably require defining a behavioral task and feature space in more concrete terms, but some intuitive/qualitative attempt at this could be valuable and increase the interest level in the field.

I gather that the manuscript overall is written intentionally to be as general as possible, but in the name of generality it may have missed an opportunity for more concrete biologically informed linkages/predictions that will excite experimentalists and theorists alike.

Specific/Minor Comments:

line 12 — Honestly, I’d go with ‘meowing’ :). I mean you can barely hear a purr, certainly not from very far away, and it’s usually associated with being stroked or contented in some other manner. [Oh, I see you already made this choice in Fig 1 and elsewhere—just have to change it in the abstract and line 125 for consistency.]

line 90, and throughout — Why use MSE instead of variance (more common in the field?), given that the estimates are assumed to be unbiased?

Figure 1F — can’t really see the dashed lines.

Figure 2D — why does the graph get all janky towards the bottom-right?

line 244 — what does Fisher Info have to do with metabolic cost?

line 592 — “there is evidence” - can you cite some here?

Several figures are described as comparing theory (analytical calculations?) with simulations, but I don’t recall seeing where it is explained how the simulated data are generated. Do they come from a generative process described by the equations, and if so, isn’t it guaranteed that they would match the ‘theory’ in the limit of large Nsims?

**Have the authors made all data and (if applicable) computational code underlying the findings in their manuscript fully available?**

Reviewer #1: Yes

Reviewer #2: Yes

PLOS authors have the option to publish the peer review history of their article (what does this mean?). If published, this will include your full peer review and any attached files.

Reviewer #1: No

Reviewer #2: No

Figure Files:

Data Requirements:

Reproducibility:

References:

---

## [Editor Report · Decision Letter 1]

4 Jul 2023

Dear Johnston,

We are pleased to inform you that your manuscript 'Redundant representations are required to disambiguate simultaneously presented complex stimuli' has been provisionally accepted for publication in PLOS Computational Biology.

Best regards,

Tianming Yang

Academic Editor

PLOS Computational Biology

Lyle Graham

Section Editor

PLOS Computational Biology

---

## [Editor Report · Acceptance letter]

2 Aug 2023

PCOMPBIOL-D-23-00239R1 

Redundant representations are required to disambiguate simultaneously presented complex stimuli

Dear Dr Johnston,

I am pleased to inform you that your manuscript has been formally accepted for publication in PLOS Computational Biology. Your manuscript is now with our production department and you will be notified of the publication date in due course.

With kind regards,

Timea Kemeri-Szekernyes
